# Gender Differences in the Severity of Cadmium Nephropathy

**DOI:** 10.3390/toxics11070616

**Published:** 2023-07-15

**Authors:** Supabhorn Yimthiang, David A. Vesey, Glenda C. Gobe, Phisit Pouyfung, Tanaporn Khamphaya, Soisungwan Satarug

**Affiliations:** 1Occupational Health and Safety, School of Public Health, Walailak University, Nakhon Si Thammarat 80160, Thailand; ksupapor@mail.wu.ac.th (S.Y.); phisit.po@mail.wu.ac.th (P.P.); tanaporn.kh@mail.wu.ac.th (T.K.); 2The Centre for Kidney Disease Research, Translational Research Institute, Brisbane 4102, Australia; david.vesey@health.qld.gov.au (D.A.V.); g.gobe@uq.edu.au (G.C.G.); 3Department of Kidney and Transplant Services, Princess Alexandra Hospital, Brisbane 4102, Australia; 4School of Biomedical Sciences, The University of Queensland, Brisbane 4072, Australia; 5NHMRC Centre of Research Excellence for CKD QLD, UQ Health Sciences, Royal Brisbane and Women’s Hospital, Brisbane 4029, Australia

**Keywords:** β_2_-microglobulin, cadmium, diabetes, GFR, hypertension, smoking, tubular proteinuria

## Abstract

The excretion of β_2_-microglobulin (β_2_M) above 300 µg/g creatinine, termed tubulopathy, was regarded as the critical effect of chronic exposure to the metal pollutant cadmium (Cd). However, current evidence suggests that Cd may induce nephron atrophy, resulting in a reduction in the estimated glomerular filtration rate (eGFR) below 60 mL/min/1.73 m^2^. Herein, these pathologies were investigated in relation to Cd exposure, smoking, diabetes, and hypertension. The data were collected from 448 residents of Cd-polluted and non-polluted regions of Thailand. The body burden of Cd, indicated by the mean Cd excretion (E_Cd_), normalized to creatinine clearance (C_cr_) as (E_Cd_/C_cr_) × 100 in women and men did not differ (3.21 vs. 3.12 µg/L filtrate). After adjustment of the confounding factors, the prevalence odds ratio (POR) for tubulopathy and a reduced eGFR were increased by 1.9-fold and 3.2-fold for every 10-fold rise in the Cd body burden. In women only, a dose–effect relationship was seen between β_2_M excretion (E_β2M_/C_cr_) and E_Cd_/C_cr_ (*F* = 3.431, η^2^ 0.021). In men, E_β2M_/C_cr_ was associated with diabetes (β = 0.279). In both genders, the eGFR was inversely associated with E_β2M_/C_cr_. The respective covariate-adjusted mean eGFR values were 16.5 and 12.3 mL/min/1.73 m^2^ lower in women and men who had severe tubulopathy ((E_β2M_/C_cr_) × 100 ≥ 1000 µg/L filtrate). These findings indicate that women were particularly susceptible to the nephrotoxicity of Cd, and that the increment of E_β2M_/C_cr_ could be attributable mostly to Cd-induced impairment in the tubular reabsorption of the protein together with Cd-induced nephron loss, which is evident from an inverse relationship between E_β2M_/C_cr_ and the eGFR.

## 1. Introduction

Cadmium (Cd) is a toxic metal pollutant that preferentially accumulates in the proximal tubule of kidneys, where it causes tubular cell injury, cell death, nephron atrophy, and eventually, a reduction in the estimated glomerular filtration rate (eGFR) below 60 mL/min/1.73 m^2^ [1,2,3,4]. As exposure to Cd in the diet is inevitable for most populations, it has become an environmental toxicant of significant worldwide public health concern. A total diet study undertaken in Japan between 2013 and 2018 reported that rice and its products, green vegetables, and cereals and seeds plus potatoes constituted 38%, 17%, and 11% of total dietary exposure, respectively [5].

To safeguard against excessive exposure to Cd in the human diet, guidelines, referred to as a tolerable intake level of Cd, were created by the Joint FAO/WHO Expert Committee on Food Additives and Contaminants (JECFA) [6]. Notably, the “tolerable” intake level was based on the risk assessment model that assumed tubular proteinuria, reflected by an increase in the excretion of the low-molecular-weight protein β2-microglubulin (β_2_M, E_β2M_) above 300 μg/g creatinine, to be an early warning sign of the nephrotoxicity of Cd. Consequently, tubulopathy is the most frequently reported adverse effect of Cd exposure. Numerous studies, however, have cast considerable doubt on the utility of E_β2M_ for such purposes.

This study aims to examine whether the exposure to Cd adversely impacts kidney toxicity differently in men and women. To this end, tubular dysfunction and changes in the eGFR were quantified in residents of Cd-polluted and non-polluted regions of Thailand and analyzed in relation to Cd exposure levels. The confounding impact of smoking, diabetes, and hypertension were also evaluated. The exposure to Cd was assessed via the measurement of blood Cd concentration ([Cd]_b_) and urinary Cd excretion (E_Cd_). Tubular dysfunction was assessed via E_β2M_. The equations developed by the Chronic Kidney Disease Epidemiology Collaboration (CKD-EPI) were used to compute the estimated GFR (eGFR) [7].

## 2. Materials and Methods

### 2.1. Participants

To obtain a group with a wide range of environmental Cd exposure amenable to dose–effect relationship assessment, we assembled data from 334 women and 114 men who participated in the cross-sectional studies conducted in a high-exposure area of the Mae Sot district, Tak province [8], and a low-exposure locality in Pakpoon Municipality of Nakhon Si Thammarat Province [9]. Based on the data from a nationwide survey of Cd levels in soils and food crops [10], environmental exposure to Cd in Nakhon Si Thammarat was low.

The study protocol for the Mae Sot group was approved by the Institutional Ethical Committees of Chiang Mai University and the Mae Sot Hospital [8]. The study protocol for the Nakhon Si Thammarat group was approved by the Office of the Human Research Ethics Committee of Walailak University in Thailand [9].

All participants gave informed consent prior to participation. They had been living at their current addresses for at least 30 years. Exclusion criteria were pregnancy, breast-feeding, a history of metal work, and a hospital record or physician’s diagnosis of an advanced chronic disease. Diabetes was defined as having fasting plasma glucose levels ≥ 126 mg/dL (https://www.cdc.gov/diabetes/basics/getting-tested.html (accessed on 25 June 2023)) or a physician’s prescription of anti-diabetic medications. Hypertension was defined as systolic blood pressure ≥ 140 mmHg, diastolic blood pressure ≥ 90 mmHg [11], a physician’s diagnosis, or prescription of anti-hypertensive medications.

### 2.2. Collection and Analysis of Blood and Urine Samples

Second morning urine samples were collected after an overnight fast, and whole blood samples were obtained within 3 h after the urine sampling. Aliquots of urine, whole blood, and plasma were stored at −20 or −80 °C prior to analysis. The assay for urine and plasma concentrations of creatinine ([cr]_u_ and [cr]_p_) was based on the Jaffe reaction. The assay of urinary β_2_M concentration ([β_2_M]_u_) was based on the latex immunoagglutination method (LX test, Eiken 2MGII; Eiken and Shionogi Co., Tokyo, Japan) or the ELISA method (Sino Biological Inc., Wayne, PA, USA).

Urinary Cd concentrations ([Cd]_u_) were determined using an atomic absorption spectrophotometer. Urine standard reference material No. 2670 (National Institute of Standards, Washington, DC, USA) or the reference urine metal control levels 1, 2, and 3 (Lyphocheck, Bio-Rad, Hercules, CA, USA) were used for quality control, analytical accuracy, and precision assurance. The limit of detection (LOD) of Cd quantitation was defined as 3 times the standard deviation of blank measurements. When [Cd]_u_ was below its detection limit (0.1 µg/L), the Cd concentration assigned was the LOD divided by the square root of 2 [12].

### 2.3. Estimated Glomerular Filtration Rates (eGFRs)

The GFR is the product of the nephron number and mean single nephron GFR, and in theory, the GFR is indicative of nephron function [13,14,15]. In practice, the GFR is estimated from established chronic kidney disease epidemiology collaboration (CKD-EPI) equations and is reported as the eGFR [15].

Male eGFR = 141 × [plasma creatinine/0.9]^Y^ × 0.993^age^, where Y = −0.411 if [cr]_p_ ≤ 0.9 mg/dL, and Y = −1.209 if [cr]_p_ > 0.9 mg/dL. Female eGFR = 144 × [plasma creatinine/0.7]^Y^ × 0.993^age^, where Y = −0.329 if [cr]_p_ ≤ 0.7 mg/dL, and Y = −1.209 if [cr]_p_ > 0.7 mg/dL. CKD stages 1, 2, 3a, 3b, 4, and 5 corresponded to eGFRs of 90–119, 60–89, 45–59, 30−44, 15–29, and <15 mL/min/1.73 m^2^, respectively.

### 2.4. Normalization of Excretion Rate

E_x_ was normalized to E_cr_ as [x]_u_/[cr]_u_, where x = Cd or β_2_M; [x]_u_ = urine concentration of x (mass/volume); and [cr]_u_ = urine creatinine concentration (mg/dL). The ratio [x]_u_/[cr]_u_ was expressed in μg/g of creatinine.

E_x_ was normalized to C_cr_ as E_x_/C_cr_ = [x]_u_[cr]_p_/[cr]_u_, where x = Cd or β_2_M; [x]_u_ = urine concentration of x (mass/volume); [cr]_p_ = plasma creatinine concentration (mg/dL); and [cr]_u_ = urine creatinine concentration (mg/dL). E_x_/C_cr_ was expressed as the excretion of x per volume of filtrate [7].

### 2.5. Statistical Analysis

Data were analyzed using IBM SPSS Statistics 21 (IBM Inc., New York, NY, USA). The Mann–Whitney U test was used to assess differences in mean values in women and men, and Pearson’s chi-squared test was used to assess differences in percentages. The one-sample Kolmogorov–Smirnov test was used to identify departures of continuous variables from a normal distribution, and logarithmic transformation was applied to variables that showed rightward skewing before they were subjected to parametric statistical analysis.

The multivariable logistic regression analysis was used to determine the prevalence odds ratio (POR) for categorical outcomes. Reduced eGFR was assigned when eGFR ≤ 60 mL/min/1.73 m^2^. For C_cr_-normalized data, tubular dysfunction was defined as (E_β2M_/C_cr_) × 100 ≥ 300 µg/L of filtrate. For E_cr_-normalized data, tubular dysfunction was defined as E_β2M_/E_cr_ ≥ 300 µg/g creatinine [6]. Univariate analysis of covariance via Bonferroni correction in multiple comparisons was used to obtain covariate-adjusted mean E_Cd_/C_cr_ and mean E_β2M_/C_cr_. For all tests, p-values ≤ 0.05 were considered to indicate statistical significance.

## 3. Results

### 3.1. Descriptive Characteristics of Participants

This cohort consisted of 334 women (mean age 51.5 years) and 114 men (mean age 49.9 years) (Table 1).

Of the total of 334 women, 224 and 110 were from the high- and low-exposure regions, respectively. In comparison, of the total 114 men, 84 and 30 males were from the high- and low-exposure regions, respectively.

The respective overall percentages of smoking, hypertension, diabetes, and reduced eGFR were 31.3%, 48.7%, 15.4%, and 6.9%. More than half of the men (68.4%) smoked cigarettes, while only 18.6% of women did. The % of all other ill health conditions in men and women did not differ, nor did their mean age differ.

With the exception of BMI, the mean plasma creatinine, mean urine creatinine, and mean blood Cd were all lower in women than in men. The mean eGFR, mean urine Cd, and mean β_2_M concentrations in women and men were not statistically different.

For the C_cr_-normalized data, the mean E_Cd_/C_cr_ and mean E_β2M_/C_cr_ in women and men both did not differ statistically. However, there were statistically significant differences in the % of women and men across three E_β2M_/C_cr_ groups.

For the E_cr_-normalized data, the mean E_Cd_/E_cr_ and mean E_β2M_/E_cr_ were higher in women than in men. The % of men across three E_β2M_/E_cr_ groups differed, but there was no difference in the % of women across the E_β2M_/E_cr_ groups.

### 3.2. Cadmium Exposure Characterization

Figure 1 provides scatterplots relating two Cd exposure indicators, namely the blood Cd concentration and the excretion rate of Cd, represented as E_Cd_/C_cr_.

A strong positive association between log([Cd]_b_ × 10^3^) and [log[(E_Cd_/C_cr_) × 10^5^] was evident in both women and men (Figure 1a). After the adjustment for the covariates and interactions, the Cd body burden ([log[(E_Cd_/C_cr_) × 10^5^] explained a larger proportion of the variation in the blood Cd concentrations (log([Cd]_b_ × 10^3^) in men (η^2^ = 0.407) than in women (η^2^ = 0.105) (Figure 1b).

To further address the variables/factors that may influence the blood Cd levels, we conducted multiple regression and univariate analyses of variance that incorporated age, BMI, log[(E_Cd_/C_cr_) × 10^5^], smoking, diabetes, and hypertension as independent variables. Table 2 provides the results of these analyses.

In women, higher [Cd]_b_ values were strongly associated with higher E_Cd_/C_cr_ (β = 0.619), and were moderately associated with smoking (β = 0.123) and younger age (β = −0.170). E_Cd_/C_cr_, age, and smoking explained, respectively, 36.7%, 5.7%, and 2.8% of the variation of [Cd]_b_ in women, while the interaction between diabetes and hypertension contributed to 1.6% of the [Cd]_b_ variability. In men, higher [Cd]_b_ values were strongly associated with higher E_Cd_/C_cr_ (β = 0.581), and were moderately associated with smoking (β = 0.184) and not having diabetes (β = −0.246). E_Cd_/C_cr_, smoking, and diabetes accounted, respectively, for 42%%, 5.5%, and 9.5% of the variation of [Cd]_b_ in men, while the interaction between smoking, diabetes, and hypertension contributed to 4.2% of the [Cd]_b_ variability.

### 3.3. Effects of Cadmium Exposure on β_2_M Excretion

We assessed the effects of Cd exposure on Eβ_2_M using multiple linear regression and univariate/covariance analyses, where the indicators of Cd exposure ([Cd]_b_ and E_Cd_) were incorporated as the independent variables together with age, BMI, smoking, diabetes, and hypertension (Table 3).

In all subjects, E_β2M_/C_cr_ was associated with age (β = 0.137), E_Cd_/C_cr_ (β = 0.283), and diabetes (β = 0.323). In women, the associations of E_β2M_/C_cr_ with these three independent variables were evident. In men, E_β2M_/C_cr_ only showed a significant association with diabetes (β = 0.279).

We next examined the association between the E_Cd_/C_cr_ and E_β2M_/C_cr_ with the scatterplots and the covariate-adjusted mean E_β2M_ in subjects grouped by E_Cd_/C_cr_ tertiles (Figure 2).

The relationship between E_β2M_/C_cr_ and E_Cd_/C_cr_ was weak and statistically insignificant in all subjects (Figure 1a), as well as in women and men (Figure 2c). However, with the adjustment for the covariates that included age and BMI, diabetes, hypertension, and smoking (Figure 2b), a significant contribution of the Cd body burden to the variability of E_β2M_ became evident when all subjects were included in an analysis (*F* = 4.473, η^2^ 0.012, *p* = 0.021). The E_β2M_ in the subjects of the high E_Cd_/C_cr_ tertile was higher compared with those of the middle and low E_Cd_/C_cr_ tertiles (Figure 1b). In the subgroup analysis (Figure 1d), a dose–effect relationship of E_Cd_ and E_β2M_ was seen in women only (*F* = 3.431, η^2^ 0.021, *p* = 0.034).

### 3.4. Effects of Cadmium Exposure on the Prevalence Odds of Tubulopathy

Table 4 provides the results of the logistic regression analysis of abnormal E_β2M_ that incorporated age, BMI, log[(E_Cd_/C_cr_) × 10^5^], gender, smoking, diabetes, and hypertension as independent variables.

Among seven independent variables, the prevalence odds ratios (POR) for (E_β2M_/C_cr_) × 100 ≥ 300–999 and ≥1000 µg/L filtrate were increased with age, log[(E_Cd_/C_cr_) × 10^5^], and diabetes. All of the other four independent variables did not show a significant association with abnormal β_2_M excretion. For every 10-fold rise in E_Cd_/C_cr_, the POR for (E_β2M_/C_cr_) × 100 of ≥300 and ≥1000 µg/L were increased by 1.94-fold and 3.34-fold, respectively.

### 3.5. Effects of Cadmium Exposure on eGFR

Similarly, we assessed the effects of Cd exposure on the estimated glomerular filtration rate (eGFR) via multiple linear regression and logistic regression analyses, where [Cd]_b_ and E_Cd_ were incorporated as the independent variables together with age, BMI, smoking, diabetes, and hypertension (Table 5).

In all subjects, the eGFR was inversely associated with age (β = −0.517), E_Cd_ (β = −0.148), and diabetes (β = −0.109). In the subgroup analysis, inverse associations of the eGFR with these three independent variables (age, E_Cd_, and diabetes) were seen only in women. In men, the eGFR was not associated with E_Cd_, but this parameter showed inverse associations with age (β = −0.506) and hypertension (β = −0.212).

In the logistic regression of a reduced eGFR (eGFR ≤ 60 mL/min/1.73 m^2^), age, BMI, log[(E_Cd_/C_cr_) × 10^5^], gender, smoking, diabetes, and hypertension were incorporated as independent variables (Table 6).

The POR values for a reduced eGFR were increased with age, log[(E_Cd_/C_cr_) × 10^5^], and diabetes. For every 10-fold rise in E_Cd_/C_cr_, the POR for a reduced eGFR was increased by 3.2-fold. There was a 4.2-fold increase in the POR for a reduced eGFR among those with diabetes.

### 3.6. Inverse Relationship of β_2_M Excretion and eGFR

Figure 3 provides scatterplots relating the eGFR to E_β2M_ among the study subjects together with the covariate-adjusted means of the eGFR in women and men.

A statistically significant inverse relationship between the eGFR and E_β2M_ was seen in all subjects (Figure 3a), as well as in women and men (Figure 3c). In all subjects (Figure 3b), the eGFR explained 5.7% of the variation in E_β2M_/C_cr_ (*F* = 12.247, *p* < 0.001).

For simplicity, the degree of tubulopathy, assessed via E_β2M,_ was graded into three levels, where levels 1, 2, and 3 of tubulopathy corresponded to (E_β2M_/C_cr_) × 100 < 300, 300–999, and ≥1000 µg/L filtrate, respectively.

The covariate-adjusted mean eGFR was 14.0 and 10.5 mL/min/1.73 m^2^ lower in subjects with level 3 tubulopathy, compared to those with tubular dysfunction levels 2 and 1, respectively (Figure 3b).

In the subgroup analysis (Figure 3d), the η^2^ values indicated a nearly twice larger effect size of the eGFR on E_β2M_ in women (η^2^ = 0.114), compared to men (η^2^ = 0.066). In women, those with level 3 tubulopathy had a covariate-adjusted mean eGFR 16.5 and 12.0 mL/min/1.73 m^2^ lower compared to those with levels 1 and 2 tubulopathy, respectively. In men, those with level 3 tubulopathy had a covariate-adjusted mean eGFR 12.3 mL/min/1.73 m^2^ lower compared to those with level 1 tubulopathy. The adjusted mean eGFR values in men with tubulopathy levels 3 and 2 did not differ statistically.

In another logistic regression, a relative contribution of Cd exposure and tubular dysfunction levels to the prevalence of a reduced eGFR was determined. E_Cd_ was entered as a continuous variable, while E_β2M_ was categorized into levels 1, 2, and 3, as previously stated. Table 7 provides the results of such analysis.

The POR values for a reduced eGFR rose with age (POR = 1.14), E_Cd_/C_cr_ (POR = 2.25), tubulopathy level 2 (POR = 8.31), and tubulopathy level 3 (POR = 33.7). All other independent variables, such as diabetes and hypertension, did not show a significant association with the POR for a reduced eGFR.

An equivalent logistic regression was conducted using E_cr_-normalized E_Cd_ and E_β2M_ data (Table 8).

The POR values for a reduced eGFR rose with age (POR = 1.10), the severity of tubular dysfunction, E_β2M_/E_cr_ 300–999 µg/g creatinine (POR = 3.20), and E_β2M_/E_cr_ ≥ 100 µg/g creatinine (POR = 19.0). Associations of POR for a reduced eGFR with E_Cd_/E_cr_ and all other variables were statistically insignificant.

## 4. Discussion

This study used a cross-sectional analysis of kidney dysfunction, tubular proteinuria, and eGFR decline to determine the differential impact of Cd exposure in men and women. Whereas many previous studies focused primarily on Cd-induced tubulopathy in women, we investigated these health outcomes in both men and women along with confounding risk factors, smoking, diabetes, and hypertension. The excretion rate of Cd and β_2_M (E_Cd_ and E_β2M_) were normalized to the surrogate measure of the GFR, creatinine clearance (C_cr_). This C_cr_-normalization of E_Cd_ and E_β2M_ as E_Cd_/C_cr_ and E_β2M_/C_cr_ depicts the excretion rates per functional nephron; thereby, it corrects for differences in the number of functioning nephrons among the study subjects [7]. This C_cr_-normalized excretion rate also corrects for urine dilution, but it is unaffected by creatinine excretion (E_cr_). Thus, E_Cd_/C_cr_ and E_β2M_/C_cr_ provide an accurate quantification of the kidney burden of Cd and its toxicity to kidney tubular cells.

We selected subjects from two population-based studies, undertaken in an area with endemic Cd contamination in the Mae Sot district, Tak province [8], and in a control, non-contaminated area in the Nakhon-Si-Thammarat province of Thailand [9,10]. The Cd content of the paddy soil samples from the Mae Sot district exceeded the standard of 0.15 mg/kg, and the rice samples collected from households contained four times the amount of the permissible Cd level of 0.1 mg/kg [16].

### 4.1. Exposure Levels of Cadmium in Women Versus Men

Men and women in this cohort carry the same body burden of Cd, which is evident from a nearly identical mean (E_Cd_/C_cr_) × 100 values of 3.12 vs. 3.21 µg/L filtrate. The sources of Cd could be differentiated through an analysis of blood–urine Cd relationships.

[Cd]_b_ and E_Cd_/C_cr_ correlated strongly with each other in women (R^2^ = 0.624) and men (R^2^ = 0.661) (Figure 1a), and the covariate-adjusted means [Cd]_b_ showed a stepwise increase through the E_Cd_/C_cr_ tertiles in both genders. Notably, E_Cd_/C_cr_ explained a larger fraction of the variation in [Cd]_b_ in men than it did in women (η^2^ 0.407 vs. 0.105) (Figure 1b). The variability in [Cd]_b_ was associated mostly with E_Cd_/C_cr_ in both genders, while smoking explained a larger fraction of the [Cd]_b_ variability among men than among women (5.5% vs. 2.8%). This result was expected, given the high % of smokers in the male group (68.4% vs. 18.6%) and the higher mean [Cd]_b_ in men than in women (3.25 vs. 4.36 µg/L). In men only, the [Cd]_b_ variation was associated with diabetes.

### 4.2. The Toxic Manifestation of Cadmium Exposure in Women Versus Men

An independent health survey reported that the prevalence of chronic kidney disease (CKD), defined as the eGFR ≤ 60 mL/min/1.73 m^2^, among Mae Sot residents was 16.1%, while the prevalence of tubulopathy, referred to as tubular proteinuria, was 36.1% [17]. This reported tubular proteinuria was based on the cut-off value of E_β2M_/E_cr_ at 300 µg/g creatinine [6], which equates to E_β2M_/C_cr_ of 2–3 µg/ L filtrate or (E_β2M_/C_cr_) × 100 of 200–300 µg/ L filtrate. Notably, the cut-off value for E_β2M_/E_cr_ at 300 µg/g creatinine was used as the critical effect of exposure to Cd in the human diet [6].

In this cohort, tubular proteinuria affected more than half of women (61.1%) and men (52.6%). One of four women had severe tubular impairment [(E_β2M_/C_cr_) × 100 ≥ 1000 µg/L filtrate], whereas one of five men had this abnormality.

In women, E_β2M_/C_cr_ showed a moderate positive association with age (β = 0.131), and an equally strong association with E_Cd_/C_cr_ (β = 0.306) and diabetes (β = 0.349) (Table 3). In men, E_β2M_/C_cr_ did not show a significant association with age or E_Cd_/C_cr_, but this tubular defect was associated with diabetes only (β = 0.279). In the covariance analysis, the contribution of E_Cd_/C_cr_ to the variability of E_β2M_/C_cr_ in women was demonstrable together with a dose–effect relationship after the adjustment of the covariates and interactions (Figure 2d). In contrast, the contribution of E_Cd_/C_cr_ to the variation of E_β2M_/C_cr_ in men was statistically insignificant (Figure 2d). An association of the marker of tubular dysfunction (E_βM_) and diabetes seen in both men and women is in line with the current knowledge that diabetes adversely affects both glomerular (GFR) and tubular function, termed diabetic tubulopathy [18,19].

The overall mean eGFR was 90 mL/min/1.73 m^2^, and the overall prevalence of eGFR ≤ 60 mL/min/1.73 m^2^ in this cohort was 6.9% (Table 1). The difference in the % of the reduced eGFR in women and men (8.1% vs. 3.5%) did not reach a statistical significance level (*p* = 0.097), nor did the difference in the mean eGFR in women and men (*p* = 0.145). The weaker effect of Cd exposure on the eGFR in men, compared to women, remains to be confirmed with a sufficiently large sample group of men. However, the regression analysis also indicated gender differences in susceptibility to the nephrotoxicity of Cd (Table 5). In women, the eGFR was inversely associated with age (β = −0.511), E_Cd_/C_cr_ (β = −0.185), and diabetes (β = −0.128). In comparison, the eGFR in men was not associated with E_Cd_/C_cr_, while showing an inverse association with age (β = −0.506) and hypertension (β = −0.212). Adverse effects of hypertension and diabetes on the eGFR have been noted in a cross-sectional study of the general U.S. population, where a Cd-induced GFR reduction was more pronounced in those who had diabetes and/or hypertension [20].

We speculate that gender differences in the levels of some protective factors, notably body status of nutritionally essential metals such as iron and zinc, may contribute to the increased susceptibility to Cd nephrotoxicity that was seen in women. Similarly, environmental Cd exposure has been linked to a reduction in the eGFR among participants in various cycles of the U.S. National Health and Nutrition Examination Survey (NHANES) undertaken over 18 years (1999 to 2016) [20,21,22]. Lin et al. (2014) reported that the risk for a reduced eGFR was higher in those with lower serum zinc (OR 3.38) compared to those with similar Cd exposure levels and serum zinc > 74 μg/dL (OR 2.04) [22].

### 4.3. Increment of β_2_M Excretion as GFR Falls

The protein β_2_M with the molecular weight of 11,800 Da is filtered freely by the glomeruli and is reabsorbed almost completely by the kidney’s tubular epithelial cells [13]. Thus, the defective tubular re-absorption of β_2_M will result in an enhanced excretion rate of β_2_M [23,24,25,26]. The loss of nephrons also raises the excretion of β_2_M for the following reasons [23,26]. When the reabsorption rate of β_2_M per nephron remains constant, its excretion will vary directly with its production. If the production and reabsorption per nephron remain constant as nephrons are lost, the excretion of β_2_M will rise [27].

It can thus be expected that the excretion of β_2_M will increase when the GFR falls for any causes. Indeed, E_β2M_/C_cr_ was inversely associated with the eGFR in both women and men (Figure 3c), although the causes of their eGFR decreases seemed to be different. In a quantitative analysis (Figure 3d), the η^2^ value describing the effect size of E_β2M_/C_cr_ on the eGFR variability was 1.7-fold larger in women than in men (0.114 vs. 0.066). In both women and men, the eGFR was the lowest in those who had (E_β2M_/C_cr_) × 100 ≥ 1000 µg/L filtrate, indicative of severely impaired tubular function.

Notably, the POR for a reduced eGFR was increased by 8.3-fold and 33.7-fold in those with (E_β2M_/C_cr_) × 100 of 300–999 and ≥1000 µg/L filtrate, respectively (Table 7), compared to those with E_β2M_/C_cr_) × 100 < 300 µg/L filtrate. A substantial loss of nephron function was a likely cause of the massive increases in E_β2M_/C_cr_ that was seen in those who had an eGFR below 60 mL/min/1.73 m^2^.

### 4.4. The Pitfall of Adjusting Excretion Rate to Ecr and Implication for Health Risk Estimation

The C_cr_-normalized data indicate that women and men shared the same burden of Cd (Table 1). The data also indicate that the % of women and men across the three categories of tubulopathy were all statistically different, thereby linking Cd exposure to the severity of tubulopathy in both genders. The logistic regression data (Table 7) show that the likelihood of having a reduced eGFR was increased by 8.3-fold and 33.7-fold in those who had (E_β2M_/C_cr_) × 100 of 300–999 and ≥1000 µg/L filtrate compared to those with (E_β2M_/C_cr_) × 100 < 300 µg/L filtrate.

The E_cr_-normalized data indicated that the mean E_Cd_/E_cr_ in women was statistically higher than that of men (4.26 vs. 3.30 µg/g creatinine). They also indicated that the difference in % of women across the three tubulopathy categories was minuscule, and that the % distribution of men across the tubulopathy categories was statistically significant. These data suggest an association of Cd exposure and the severity of tubulopathy in men only. The logistic regression data (Table 8) show that the likelihood of having a reduced eGFR was increased by 3.2-fold and 19-fold in those who had E_β2M_/E_cr_ of 300–999 and ≥1000 µg/g creatinine compared to those with E_β2M_/E_cr_ < E_β2M_/E_cr_.

Previously, E_β2M_/E_cr_ of 100–299, 300–999, and ≥1000 μg/g creatinine were found to be associated with 4.7-fold, 6.2-fold, and 10.5-fold increases in the prevalence odds of a reduced eGFR [28,29]. Similarly, a rise in E_β2M_/E_cr_ to levels not higher than 100 μg/g creatinine was associated with an increased risk of hypertension in the general Japanese population [30], while the prospective cohort data showed that E_β2M_/E_cr_ was associated with a 79% increase in the likelihood of having a large decline in the eGFR (10 mL/min/1.73 m^2^) over a five-year period [31]. Thus, a cut-off value for E_β2M_/E_cr_ above 300 μg/g creatinine does not reflect an early warning sign of the nephrotoxicity of Cd. The utility of this E_β2M_/E_cr_ value as a toxicity criterion to derive a toxicity threshold level for Cd is inappropriate.

In summary, adjusting E_Cd_ and E_β2M_ to E_cr_ produces an erroneous interpretation of the effect of Cd exposure on the eGFR, while underestimating the severity of Cd-induced tubulopathy, especially among women. These data call into question the utility of E_β2M_/E_cr_ of 300 µg/g creatinine to represent the critical effect of exposure to Cd in the human diet. New health guidance values need to be established for this toxic metal, and new public measures are needed to minimize the Cd contamination of food chains.

### 4.5. Strength and Limitation

In this cohort, the levels of environmental exposure among the participants were assessed by measuring the blood Cd and urinary Cd excretion rates. Strong correlations between these two parameters were seen in both men and women (Figure 1). Both the tubular and glomerular function were examined concurrently together with confounding factors, smoking, hypertension, and type 2 diabetes. These are the strengths of our study.

The small number of males from high- (*n* = 84) and low-exposure (*n* = 30) locations is a limitation. This precludes an analysis of both genders separately from both locations that may help to rule out any other environmental effects on adverse kidney outcomes in women. In addition, the heterogeneity in the hormonal status, notably estrogen, in female participants who are menopausal and post-menopausal is a limitation [32].

## 5. Conclusions

The excretion of β_2_M above 300 µg/g creatinine (≈2–3 µg/L filtrate) and a reduction in the glomerular function, indicated by an eGFR below 60 mL/min/1.73 m^2^, are the manifestations of severe kidney toxicities due to chronic exposure to Cd that are more prevalent and more severe in women than men of the same body burden.

## Figures and Tables

**Figure 1 toxics-11-00616-f001:**
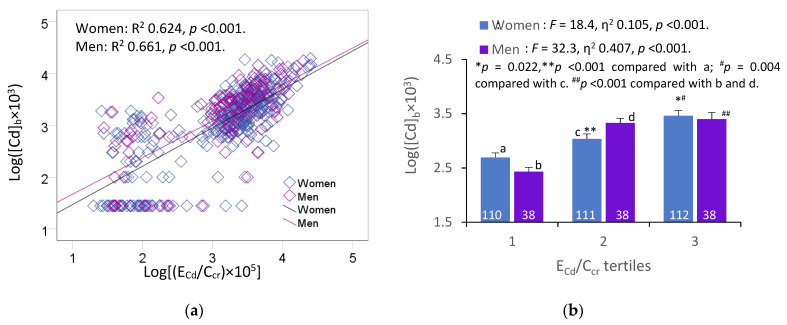
Blood cadmium and urinary cadmium excretion relationship. Scatterplots relating log([Cd]_b_ × 10^3^] to log[(E_Cd_/C_cr_) × 10^5^] in women and men (**a**). Coefficients of determination (R^2^) and *p*-values are provided for all scatterplots. Bar graph (**b**) depicts mean log([Cd]_b_ × 10^3^] values in women and men across E_Cd_/C_cr_ tertiles. Letters a and c refer to groups of women whose E_Cd_/C_c_r values were in low and middle E_Cd_/C_cr_ tertiles, respectively. Letters b and d refer to groups of men whose E_Cd_/C_cr_ values were in low and middle E_Cd_/C_cr_ tertiles, respectively. All means were obtained via univariate analysis with adjustment for covariates and interactions. For women, respective arithmetic means and standard deviations (SD) for (E_Cd_/C_cr_) × 100 tertiles 1, 2, and 3 are 0.37 (0.47), 2.34 (0.52), and 6.84 (4.46) µg/L of filtrate. For men, respective arithmetic means and standard deviations (SD) for (E_Cd_/C_cr_) × 100 tertiles quartiles 1, 2, and 3 are 0.36 (0.42), 2.14 (0.63), and 6.81 (3.78) µg/L of filtrate. For all tests, *p*-values ≤ 0.05 indicate statistically significant differences.

**Figure 2 toxics-11-00616-f002:**
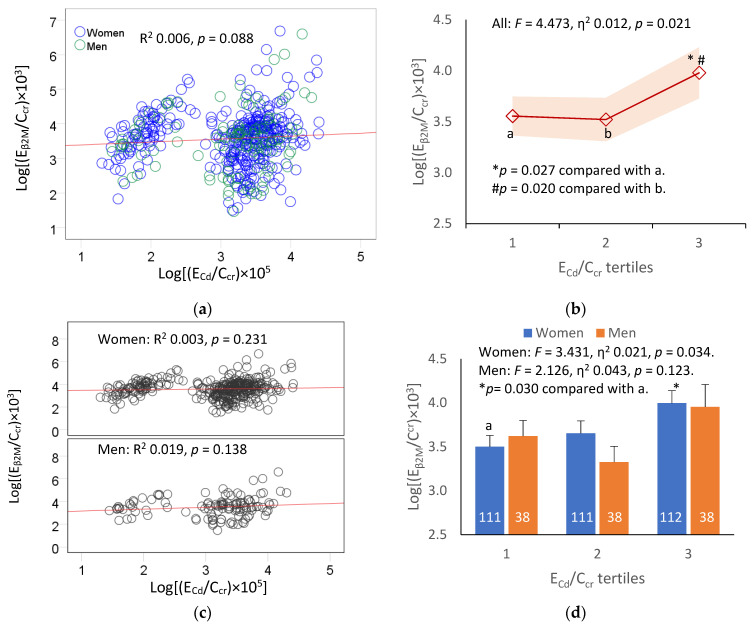
Dose–effect relationship of β_2_-microgloubulin and cadmium excretion. Scatterplots relate log[(E_βM_/C_cr_) × 10^3^] to log[(E_Cd_/C_cr_) × 10^5^] in all subjects (**a**), and in women and men (**c**). Coefficients of determination (R^2^) and *p*-values are provided for all scatterplots. The color-coded area graph (**b**) depicts means of log[(E_β2M_/C_cr_) × 10^3^] across E_Cd_/C_cr_ tertiles. Shaded areas indicate variability of means. Bar graph (**d**) depicts mean log[(E_β2M_/C_cr_) × 10^3^] in women and men in each E_Cd_/C_cr_ tertile. The respective numbers of subjects in E_Cd_/C_cr_ tertiles 1, 2, and 3 are 149, 149, and 150. In Figure 2d, a letter a refers to a group of women whose ECd/C_cr_ values were in low E_Cd_/C_cr_ tertile. All means were obtained via univariate analysis with adjustment for covariates and interaction. For women, respective arithmetic means and standard deviations (SD) for (E_Cd_/C_cr_) × 100 tertiles 1, 2, and 3 are 0.37 (0.47), 2.34 (0.52), and 6.84 (4.46) µg/L of filtrate. For men, respective arithmetic means and standard deviations (SD) for (E_Cd_/C_cr_) × 100 tertiles 1, 2, and 3 are 0.36 (0.42), 2.14 (0.63), and 6.81 (3.78) µg/L of filtrate. For all tests, *p*-values ≤ 0.05 indicate statistically significant differences.

**Figure 3 toxics-11-00616-f003:**
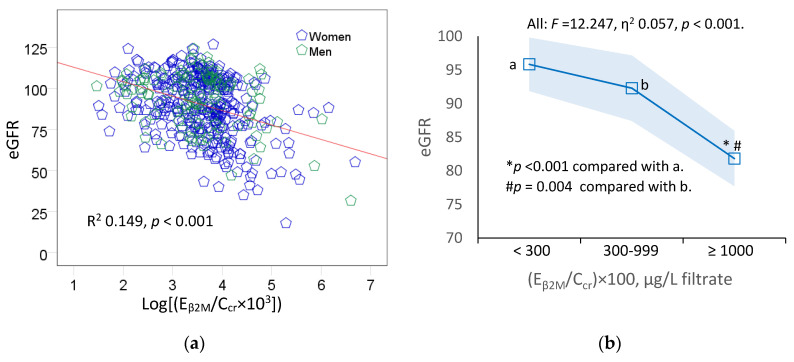
An inverse relationship of eGFR with β_2_-microgloubulin excretion. Scatterplots relate eGFR to log[(E_βM_/C_cr_) × 10^3^] in all subjects (**a**), and in women and men (**c**). Coefficients of determination (R^2^) and *p*-values are provided for all scatterplots. The color-coded area graph (**b**) depicts means of eGFR across three E_β2M_/C_cr_ ranges. Shaded areas indicate variability of means. Bar graph (**d**) depicts means of eGFR in women and men in each E_β2M_/C_cr_ range. The respective numbers of subjects in (E_β2M_/C_cr_) × 100 < 300, 300−999 and ≥1000 µg/L of filtrate are 184, 156, and 109. In Figure 3d, letters a and c refer to groups of women whose (E_β2M_/C_cr_) × 100 < 300 and 300–999 µg/L filtrate, respectively. The letter b refers to a group of men whose (E_β2M_/C_cr_) × 100 < 300 µg/L filtrate, respectively. All means were obtained via univariate analysis with adjustment for covariates and interaction. For all tests, *p*-values ≤ 0.05 indicate statistically significant differences.

**Table 1 toxics-11-00616-t001:** Characteristics of study subjects.

Parameters	All Subjects, *n* = 448	Women, *n* = 334	Men, *n* = 114	*p*
Age, years	51.1 ± 8.6	51.5 ± 9.0	49.9 ± 7.2	0.344
BMI, kg/m^2^	24.8 ± 4.0	25.2 ± 4.0	23.7 ± 3.6	<0.001
Smoking, %	31.3	18.6	68.4	<0.001
Hypertension, %	48.7	50.6	43.0	0.160
Diabetes, %	15.4	16.2	13.2	0.442
eGFR ^a^, mL/min/1.73 m^2^	90 ± 18	89 ± 19	93 ± 16	0.145
Reduced eGFR ^b^, %	6.9	8.1	3.5	0.097
Plasma creatinine, mg/dL	0.82 ± 0.22	0.78 ± 0.21	0.95 ± 0.21	<0.001
Urine creatinine, mg/dL	114 ± 74	108 ± 74	132 ± 72	<0.001
Blood Cd, µg/L	2.75 ± 3.19	2.58 ± 3.10	3.25 ± 3.41	0.038
Urine Cd, µg/L	4.22 ± 5.67	4.36 ± 6.14	3.82 ± 4.01	0.875
Urine β_2_M, µg/L	3122 ± 18,836	2596 ± 17,238	4665 ± 22,903	0.544
Normalized to C_cr_ (E_x_/C_cr_) ^c^				
(E_Cd_/C_cr_) × 100, µg/L filtrate	3.19 ± 3.72	3.21 ± 3.79	3.12 ± 3.55	0.639
(E_β2M_/C_cr_) × 100, µg/L filtrate	3839 ± 30,422	3078 ± 26,986	6072 ± 38,837	0.212
(E_β2M_/C_cr_) × 100, µg/L filtrate, %				
<300	41.1	38.9	47.4	
300−999	34.8	35.9	31.6	
≥1000	24.1	25.1 *	21.1 ***	
Normalized to E_cr_ (E_x_/E_cr_) ^d^				
E_Cd_/E_cr_, µg/g creatinine	4.02 ± 4.41	4.26 ± 4.62	3.30 ± 3.68	0.028
E_β2M_/E_cr_, µg/g creatinine	3220 ± 21,847	3005 ± 22,812	3850 ± 18,815	0.017
E_β2M_/E_cr_, µg/g creatinine, %				
<300	35.5	32.3	45.6	
300−999	34.6	34.7	34.2	
≥1000	29.7	32.9	20.2 **	

*n*, number of subjects; BMI, body mass index; eGFR, estimated glomerular filtration rate; β_2_M, β_2_-microglobulin; E_x_, excretion of x; cr, creatinine; C_cr_, creatinine clearance; Cd, cadmium; ^a^ eGFR was determined by established CKD-EPI equations [15]; ^b^ reduced eGFR corresponds to eGFR ≤ 60 mL/min/1.73 m^2^; ^c^ E_x_/E_cr_ = [x]_u_/[cr]_u_; ^d^ E_x_/C_cr_ = [x]_u_[cr]_p_/[cr]_u_, where x = Cd or β_2_M [7]. Data for all continuous variables are arithmetic means ± standard deviation (SD). For all tests, *p* ≤ 0.05 identifies statistical significance, determined by Pearson’s chi-squared test for % differences and by the Mann–Whitney U test for mean differences between women and men. * *p* = 0.005; ** *p* = 0.004; *** *p* = 0.002.

**Table 2 toxics-11-00616-t002:** Determinants of blood cadmium concentration in women versus men.

Independent Variables/Factors	Log([Cd]_b_ × 10^3^), µg/L					
	Women, *n* = 334			Men, *n* = 114		
	β	η^2^	*p*	β	η^2^	*p*
Age, years	−0.170	0.057	<0.001	−0.004	1 × 10^−6^	0.946
BMI, kg/m^2^	−0.019	0.001	0.586	−0.079	0.022	0.180
Log[(E_Cd_/C_cr_) × 10^5^], µg/L filtrate	0.619	0.367	<0.001	0.581	0.420	<0.001
Smoking	0.123	0.028	0.001	0.184	0.055	0.002
Diabetes	−0.053	0.010	0.182	−0.246	0.095	<0.001
Hypertension	0.048	0.007	0.162	−0.061	0.004	0.287
DM × HTN	n/a	0.016	0.023	n/a	n/a	n/a
SMK × DM × HTN	n/a	n/a	n/a	n/a	0.042	0.036
Adjusted R^2^	0.624	n/a	<0.001	0.661	n/a	<0.001

β, standardized regression coefficient; adjusted R^2^, coefficient of determination; DM, diabetes; HTN, hypertension; SMK, smoking; n/a, not applicable. β indicates strength of association of log([Cd]_b_ × 10^3^) with independent variables (first column). Adjusted R^2^ indicates a fractional variation of log([Cd]_b_ × 10^3^) explained by all independent variables. Eta square (η^2^) indicates the fraction of the variability of each dependent variable explained by a corresponding independent variable. *p*-values ≤ 0.05 indicate a statistically significant contribution of variation of an independent variable to variation of a dependent variable.

**Table 3 toxics-11-00616-t003:** Associations of β_2_-mcirogloubulin excretion with cadmium exposure measurements.

Independent Variables/Factors	Log[(E_β2M_/C_cr_) × 10^3^], µg/L Filtrate					
	All Subjects		Women		Men	
	β	*p*	β	*p*	β	*p*
Age, years	0.137	0.013	0.131	0.041	0.128	0.238
BMI, kg/m^2^	−0.089	0.065	−0.102	0.062	−0.043	0.664
Log([Cd]_b_ × 10^3^), µg/L filtrate	−0.016	0.824	−0.083	0.328	0.217	0.180
Log[(E_Cd_/C_cr_) × 10^5^], µg/L filtrate	0.283	<0.001	0.306	<0.001	0.175	0.247
Gender	0.052	0.318	−	−	−	−
Smoking	0.063	0.255	0.093	0.094	−0.065	0.526
Diabetes	0.323	<0.001	0.349	<0.001	0.279	0.017
Hypertension	0.015	0.745	−0.023	0.660	0.142	0.139
Adjusted R^2^	0.105	<0.001	0.125	<0.001	0.059	0.060

β, standardized regression coefficient; adjusted R^2^, coefficient of determination. β indicates strength of association of log[(E_β2M_/C_cr_) × 10^3^] with independent variables (first column). Adjusted R^2^ indicates a fractional variation of log[(E_β2M_/C_cr_) × 10^3^] explained by all independent variables. For each test, *p*-values ≤ 0.05 indicate a statistically significant contribution of an independent variable to log[(E_β2M_/C_cr_) × 10^3^] variability.

**Table 4 toxics-11-00616-t004:** Prevalence odds for excessive excretion of β_2_M in relation to cadmium excretion and other variables.

Independent Variables/Factors	Number of Subjects	(E_β2M_/C_cr_) × 100 ≥ 300 µg/L		(E_β2M_/C_cr_) × 100 ≥ 1000 µg/L	
		POR (95% CI)	*p*	POR (95% CI)	*p*
Age, years	448	1.036 (1.007, 1.067)	0.016	1.062 (1.027, 1.098)	<0.001
BMI, kg/m^2^	448	0.971 (0.919, 1.025)	0.284	0.958 (0.896, 1.023)	0.203
Log[(E_Cd_/C_cr_) × 10^5^], µg/L filtrate	448	1.940 (1.344, 2.802)	<0.001	3.343 (2.036, 5.488)	<0.001
Gender (F/M)	334/114	1.406 (0.835, 2.367)	0.200	1.299 (0.687, 2.458)	0.421
Smoking	140	1.067 (0.645, 1.765)	0.801	1.388 (0.763, 2.522)	0.282
Diabetes	69	5.294 (2.526, 11.09)	<0.001	11.52 (5.004, 26.50)	<0.001
Hypertension	218	1.066 (0.714, 1.592)	0.753	1.535 (0.942, 2.501)	0.085

POR, prevalence odds ratio; CI, confidence interval. The units of (E_β2M_/C_cr_) × 100 and log[(E_Cd_/C_cr_) × 10^5^] are µg/L filtrate; data were generated from logistic regression analyses, relating POR for excessive β_2_M excretion to seven independent variables (first column). For all tests, *p*-values ≤ 0.05 indicate a statistically significant association of POR with a given independent variable.

**Table 5 toxics-11-00616-t005:** Associations of eGFR with cadmium exposure measurements and other variables.

Independent Variables/Factors	eGFR, mL/min/1.73 m^2^					
	All Subjects		Women		Men	
	β	*p*	β	*p*	β	*p*
Age, years	−0.517	<0.001	−0.511	<0.001	−0.506	<0.001
BMI, kg/m^2^	−0.064	0.136	−0.048	0.327	−0.149	0.095
Log([Cd]_b_ × 10^3^), µg/L filtrate	0.053	0.420	0.102	0.182	−0.153	0.291
Log[(E_Cd_/C_cr_) × 10^5^], µg/L filtrate	−0.148	0.026	−0.185	0.018	0.011	0.933
Gender	−0.001	0.977	−	−	−	−
Smoking	0.025	0.610	0.018	0.717	0.071	0.438
Diabetes	−0.109	0.023	−0.128	0.021	−0.055	0.593
Hypertension	−0.079	0.055	−0.050	0.295	−0.212	0.014
Adjusted R^2^	0.279	<0.001	0.281	<0.001	0.249	<0.001

eGFR, estimated glomerular filtration rate; β, standardized regression coefficient; adjusted R^2^, coefficient of determination. β indicates strength of association of eGFR with independent variables (first column). Adjusted R^2^ indicates a fractional variation of eGFR explained by all independent variables. For each test, *p*-values ≤ 0.05 indicate a statistically significant contribution of an independent variable to eGFR variability.

**Table 6 toxics-11-00616-t006:** Prevalence odds for a reduced eGFR in relation to cadmium excretion and other variables.

Independent Variables/ Factors	Reduced eGFR ^a^				
	β Coefficients	POR	95% CI		*p*
	(SE)		Lower	Upper	
Age, years	0.136 (0.027)	1.146	1.086	1.209	<0.001
BMI, kg/m^2^	0.020 (0.051)	1.021	0.923	1.128	0.688
Log[(E_Cd_/C_cr_) × 10^5^], µg/L filtrate	1.154 (0.358)	3.172	1.572	6.402	0.001
Gender	−0.542 (0.649)	0.582	0.163	2.075	0.404
Smoking	−0.228 (0.583)	0.796	0.254	2.493	0.695
Diabetes	1.439 (0.524)	4.217	1.510	11.78	0.006
Hypertension	0.115 (0.427)	1.122	0.486	2.591	0.787

^a^ Reduced eGFR is defined as the estimated glomerular filtration rate ≤ 60 mL/min/1.73 m^2^; β, regression coefficient; POR, prevalence odds ratio; SE, standard error of mean; CI, confidence interval. Data were generated from logistic regression, relating POR for a reduced eGFR to seven independent variables (first column). For each test, *p*-values ≤ 0.05 indicate a statistically significant contribution of individual independent variables to the POR for a reduced eGFR.

**Table 7 toxics-11-00616-t007:** Prevalence odds of a reduced eGFR in relation to cadmium and β_2_M excretion rates normalized to C_cr_.

Independent Variables/ Factors	Reduced eGFR ^a^				
	Number of Subjects	POR	95% CI		*p*
			Lower	Upper	
Age, years	448	1.140	1.072	1.211	<0.001
BMI, kg/m^2^	448	1.066	0.950	1.198	0.278
Log[(E_Cd_/C_cr_) × 10^5^], µg/L filtrate	448	2.251	1.043	4.858	0.039
Gender (F/M)	334/114	0.674	0.176	2.583	0.565
Smoking	140	0.885	0.270	2.899	0.840
Diabetes	69	1.216	0.394	3.753	0.734
Hypertension	218	1.199	0.487	2.957	0.693
(E_β2M_/C_cr_) × 100, µg/L filtrate					
<300	184	Referent			
300–999	156	8.310	2.655	26.01	<0.001
≥1000	108	33.731	4.193	271.3	0.001

^a^ Reduced eGFR is defined as the estimated glomerular filtration rate ≤ 60 mL/min/1.73 m^2^; POR, prevalence odds ratio; CI, confidence interval. Data were generated from multivariable logistic regression analyses, relating the POR for a reduced eGFR to eight independent variables (first column). *p*-values < 0.05 indicate a statistically significant increase in the POR for a reduced eGFR.

**Table 8 toxics-11-00616-t008:** Prevalence odds of a reduced eGFR in relation to cadmium and β_2_M excretion rates normalized to Ecr.

Independent Variables/ Factors	Reduced eGFR ^a^				
	Number of Subjects	POR	95% CI		*p*
			Lower	Upper	
Age, years	448	1.101	1.045	1.160	<0.001
BMI, kg/m^2^	448	1.032	0.929	1.147	0.555
Log[(E_Cd_/E_cr_) × 10^3^], µg/g creatinine	448	1.278	0.630	2.593	0.497
Gender (F/M)	334/114	0.671	0.181	2.484	0.550
Smoking	140	0.762	0.240	2.418	0.644
Diabetes	69	1.614	0.581	4.484	0.359
Hypertension	218	1.041	0.449	2.415	0.926
(E_β2M_/C_cr_) × 100, µg/g creatinine					
<300	160				
300–999	155	3.204	1.226	8.375	0.018
≥1000	133	19.042	2.387	151.907	0.005

^a^ Reduced eGFR is defined as the estimated glomerular filtration rate ≤ 60 mL/min/1.73 m^2^; POR, prevalence odds ratio; CI, confidence interval. Data were generated from multivariable logistic regression analyses, relating the POR for a reduced eGFR to eight independent variables (first column). *p*-values < 0.05 indicate a statistically significant increase in the POR for a reduced eGFR.

## Data Availability

All data are contained within this article.

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
