# Peer review of "Gender Differences in the Severity of Cadmium Nephropathy"

_toxics, 2023, doi:10.3390/toxics11070616_

Round 1

Reviewer 1 Report

To obtain a group with a wide range of environmental Cd exposure amenable to dose-effect relationship assessment,  the authors assembled data from 334 women and 114 men who participated in the cross-sectional studies conducted in a high-exposure area and a low-exposure locality. The results showed that women were particularly susceptible to the nephrotoxicity of Cd, and the increment of Eβ2M/Ccr could be attributable mostly to Cd-induced impairment in tubular reabsorption of the protein together with  Cd-induced nephron loss. The research design is appropriate. and I think  the manuscript is  very interested to teaders. However, I cannot undstand the statistical analysis. For example, in table 1, why the authors chose #p = 0.005; * p = 0.004; **p = 0.002? In figure 1b, what is the meaning a b c d? Please check!

Author Response

Reviewer 1

Comments and Suggestions

To obtain a group with a wide range of environmental Cd exposure amenable to dose-effect relationship assessment, the authors assembled data from 334 women and 114 men who participated in the cross-sectional studies conducted in a high-exposure area and a low-exposure locality. The results showed that women were particularly susceptible to the nephrotoxicity of Cd, and the increment of Eβ2M/Ccr could be attributable mostly to Cd-induced impairment in tubular reabsorption of the protein together with Cd-induced nephron loss. The research design is appropriate. and I think the manuscript is very interesting to readers However, I cannot understand the statistical analysis. For example, in table 1, why the authors chose #p = 0.005; * p = 0.004; **p = 0.002? In figure 1b, what is the meaning a, b, c, d? Please check!

Response: 

We thank the reviewer for their evaluation of our manuscript and for the helpful comments on the statistics and figure legends. We have now made appropriate changes which should further clarify these (Table 1, line 142). Changes to in the text are in blue. Below those changes are itemized.

The statistical significance levels in Table 1 #p = 0.005; * = 0.004; **p = 0.002 have now been changed to * p = 0.005, ** p = 0.004 and ***p = 0.002, respectively.

“Letters a and c refer to groups of women whose ECd/Ccr values were in low and middle ECd/Ccr tertiles, respectively. Letters b and d refer to groups of men whose ECd/Ccr values were in low and middle ECd/Ccr tertiles, respectively.”

Reviewer 2 Report

In this manuscript, Yimthiang and colleagues test whether if Cd exposure leads to renal dysfunction differently in men and women. A total of 448 participants (334 women and 114 men) were recruited from Cd-polluted and non-polluted regions of Thailand. Using urine and blood samples, the authors quantify tubular dysfunction and changes in estimated glomerular filtration rate (eGFR) in response to Cd exposure. They evaluate the confounding impact of smoking, diabetes, and hypertension and conclude that the manifestations of kidney toxicity associated with chronic Cd exposure are more prevalent and severe in women than men, given the same overall Cd burden. This manuscript makes an important contribution to toxicology, particularly in the area of sex differences. Please see our comments below.

1.    Most of the calculated parameters show standard deviations larger than the mean. Additionally, the sample size of women is almost 3 times greater than men. The authors should acknowledge this as a limitation in their study. 

2.    Since the age of women participants is 51.5+/-9 years, this likely represents a heterogenous population: some are post-menopausal, some are undergoing menopause, and some are still cycling. Hormonal changes may affect eGFR or other kidney functions. The authors may consider this parameter as another potentially confounding factor.

3.    In table 1, the % of diabetic men (0.442) seems wrong (probably should be fraction).

4.    It is not clear why, in tables 6, 7, and 8, only male gender is included as an independent variable.

5.    In table 2, under men, I think the first column should be β not p.

6.    It is not clear how many participants (both men and women) are from the 2 different study locations. Both genders may be analyzed separately from both locations to rule out any other environmental effects on women kidney functions.

English is in general good.

Author Response

Reviewer 2

Comments and Suggestions

In this manuscript, Yimthiang and colleagues test whether if Cd exposure leads to renal dysfunction differently in men and women. A total of 448 participants (334 women and 114 men) were recruited from Cd-polluted and non-polluted regions of Thailand. Using urine and blood samples, the authors quantify tubular dysfunction and changes in estimated glomerular filtration rate (eGFR) in response to Cd exposure. They evaluate the confounding impact of smoking, diabetes, and hypertension and conclude that the manifestations of kidney toxicity associated with chronic Cd exposure are more prevalent and severe in women than men, given the same overall Cd burden. This manuscript makes an important contribution to toxicology, particularly in the area of sex differences. Please see our comments below.

Response: We thank the reviewer for comments and suggestion to improve our manuscript.  We provide below point-by-point response to issues and concerns raised.

Comment 1. Most of the calculated parameters show standard deviations larger than the mean. Additionally, the sample size of women is almost 3 times greater than men. The authors should acknowledge this as a limitation in their study

Response 1: We have included a new subsection, 4.5 Strength and limitation (lines 465-476) in the Discussion, given below. A new reference, 32 has been added.

“In this cohort, levels of environmental exposure among participants were assessed by measuring the blood Cd and urinary Cd excretion rate. Strong correlations between these two parameters were seen in both men and women (Figure 1). Both the tubular and glomerular function were examined concurrently together with confounding factors, smoking, hypertenson and diabetes type 2. These are the strengths of our study. 

The small number of males from high-(n = 84) and low-exposure (n = 30) locations is a limitation. This precludes an analysis of both genders separately from both locations that may help rule out any other environmental effects on adverse kidney outcomes in women. In addition, the heterogeneity in hormonal status, notably estrogen, in menopause and post-menopausal women participants is a limitation [32].”

32. Vahter, M.; Berglund, M.; Akesson, A. Toxic metals and the menopause. J. Br. Menopause Soc. 2004, 10, 60-64.

Comment 2.  Since the age of women participants is 51.5+/-9 years, this likely represents a heterogenous population: some are post-menopausal, some are undergoing menopause, and some are still cycling. Hormonal changes may affect eGFR or other kidney functions. The authors may consider this parameter as another potentially confounding factor.

Response 2: We have included the potential hormonal influences as a limitation. Please see response to comment 1 above.

Comment 3.   In table 1, the % of diabetic men (0.442) seems wrong (probably should be fraction).

Response 3: The % of diabetic men of 0.442 was in error and it has been corrected to 13.2 %.

Comment 4.   It is not clear why, in tables 6, 7, and 8, only male gender is included as an independent variable.

Response 4: Both male and female genders were included in the analysis shown in Tables 6, 7 and 8. We have now replaced the word “Gender (male)” with “Gender (F/M)”.

Comment 5.   In table 2, under men, I think the first column should be β not p.

Response 5: We have replaced “p” in the first column for male gender with “β”.

Comment 6.    It is not clear how many participants (both men and women) are from the 2 different study locations. Both genders may be analyzed separately from both locations to rule out any other environmental effects on women kidney functions.

Response 6: We have inserted the below statements to indicate the number of men and women from each location (lines 144-146). We have included these small number of men from each location as a limitation which precludes the analysis the Reviewers suggested (please see response to comment 1 above).  

“Of the total of 334 women, 224 and 110 were from the high- and low-exposure regions, respectively. In comparison, of the total 114 men, 84 and 30 males were from the high- and low-exposure regions, respectively.”

Comments on the Quality of English Language

English is in general good.

Response: We have checked throughout the paper for typo and minor grammatical errors.
